# Pace v0.2: A Python-based Performance-Portable Atmospheric Model

Johann Dahm[1,*], Eddie Davis[1,*], Florian Deconinck[1,*], Oliver Elbert[1,*], Rhea George[1,*],
Jeremy McGibbon[1,*], Tobias Wicky[1,*], Elynn Wu[1,*], Christopher Kung[2], Tal Ben-Nun[3], Lucas Harris[4],
Linus Groner[5], and Oliver Fuhrer[1,6]

[*]These authors contributed equally to this work and are listed in alphabetical order.
[1]Allen Institute of Artificial Intelligence, Seattle, U.S.A.
[2]Global Modeling and Assimilation Office, NASA, Greenbelt MD, U.S.A.
[3]Department of Computer Science, ETH Zurich, Zurich, Switzerland
[4]Geophysical Fluid Dynamics Laboratory, NOAA, Princeton NJ, U.S.A.
[5]Swiss National Supercomputing Centre (CSCS), ETH Zurich, Lugano, Switzerland
[6]Federal Institute of Meteorology and Climatology MeteoSwiss, Zurich, Switzerland

**Correspondence:** oliver.fuhrer@meteoswiss.ch

**Abstract.** Progress in leveraging current and emerging high-performance computing infrastructures using traditional weather and climate models has been slow. This has become known more broadly as the software productivity gap. With the end of Moore's Law driving forward rapid specialization of hardware architectures, building simulation codes on a low-level language with hardware specific optimizations is a significant risk. As a solution, we present Pace, an implementation of the nonhydrostatic FV3 dynamical core and GFDL cloud microphysics scheme which is entirely Python-based. In order to achieve high performance on a diverse set of hardware architectures, Pace is written using the GT4Py domain-specific language. We demonstrate that with this approach we can achieve portability and performance, while significantly improving the readability and maintainability of the code as compared to the Fortran reference implementation. We show that Pace can run at scale on leadership-class supercomputers and achieve performance speeds 3.5-4 times faster than the Fortran code on GPU-accelerated supercomputers. Furthermore, we demonstrate how a Python-based simulation code facilitates existing or enables entirely new use-cases and workflows. Pace demonstrates how a high-level language can insulate us from disruptive changes, provide a more productive development environment, and facilitate the integration with new technologies such as machine learning.

## 1 Introduction

Current weather and climate models are written in low-level compiled languages such as Fortran for performance (Méndez et al., 2014), and typically run on high-performance computing (HPC) systems with CPUs. With the end of Moore's law (e.g. Theis and Wong, 2017), HPC systems are increasingly relying on specialized hardware architectures such as graphics processing units (GPUs) to increase throughput while maintaining a reasonable power envelope (Strohmaier et al., 2015). This has led to a number of efforts to port existing weather and climate models to run on such heterogeneous architectures, for example by adding OpenACC directives (Lapillonne et al., 2017; Clement et al., 2019; Giorgetta et al., 2022). Today, there are

a handful of successful productive deployments of weather and climate models (e.g., COSMO, MPAS) on GPU-accelerated supercomputers. But porting large code bases using compiler directives comes at a price: The maintenance cost increases significantly due to more hardware specific details being explicitly exposed in the user code and due to the fact that conditional compilation increases code complexity and likelihood of introducing errors. Also, optimizations often need to be tailored to a hardware target which may lead to code duplication and specialized implementations. As a result, community codes have been

slow in their adoption of novel and emerging hardware architectures, and increasingly complex code bases hinder their further development. The term *software productivity gap* has been coined to describe this situation (Lawrence et al., 2018).

Alternative approaches are being explored: The Simple Cloud-Resolving E3SM Atmosphere Model (E3SM Project, 2021) is a global atmosphere model implemented in C++ using the Kokkos library (Trott et al., 2022), which provides abstractions for parallel execution and data management for a wide range of programming models and target architectures using a technique

called template meta-programming. LFric, the next generation weather and climate modeling system being developed by the UK Met Office (Adams et al., 2019), is implemented using a domain-specific language (DSL) embedded in Fortran and leverages a domain-specific compiler named PSyclone to generate exectuable parallel code. While these approaches are very promising, they have currently not been adopted more widely in the field of weather and climate science.

A compelling alternative approach has been extremely successful in the trending field of machine learning: frameworks

such as PyTorch (Paszke et al., 2019) and Tensorflow (Abadi et al., 2015) have accelerated the rapid and broad adoption of machine learning methods. These frameworks allow users to implement algorithms with an abstract high-level implementation in Python using a syntax which is reminiscent of doing computation using the NumPy library (Harris et al., 2020a). Different backends allow users to target a diverse set of hardware architectures for efficient execution while retaining a high degree of programmer productivity. The approach has been shown to scale and the computational effort to train such models (e.g. Brown

et al., 2020) is comparable in size to high-resolution weather and climate simulations.

Aside from the model code itself, scientists developing and using compiled models have increasing needs to interface model code with scripting languages online. The drastic increase in model resolution over the past decades has increased the need for online diagnostic calculations to avoid slow I/O operations. It can also simplify development of machine learning parameterizations to be able to interface Python code with a compiled model. This has motivated scientists to interface Python code with

models online, for example by calling Python from Fortran (Brenowitz and Bretherton, 2019; Partee et al., 2021a, b) and by wrapping Fortran models to be driven by Python (Monteiro et al., 2018; van den Oord et al., 2020; McGibbon et al., 2021).

We present Pace, an open-source performance-portable implementation of the FV3 dynamical core (Putman and Lin, 2007; Harris et al., 2021) and GFDL microphysics (Chen and Lin, 2013; Zhou et al., 2019) written entirely in Python. Pace uses the GridTools for Python (GT4Py) DSL which separates the definition of numerical algorithms from the specific implementation

for a given hardware architecture. Optimization details such as storage order, execution schedule, placement of data in memory hierarchy, and loop bounds are not the responsibility of the domain scientist. This allows the use of a single unified and concise codebase across hardware backends, which clearly presents numerical operators and executes efficiently. The same model code can be used for applications in the classroom inside a Jupyter notebook as well as deployment at scale on large high-performance computing systems. Using Python as the host-language enables highly productive model development, testing

and validation workflows. By having access to a large ecosystem of well-maintained Python packages entirely new workflows are enabled (see Section 6.2).

The outline of the paper is as follows: We start in Section 2 with an overview of the DSL structure before moving on to Pace itself in Section 3. In Section 4 we discuss the process of porting and validating the code from Fortran to Python in detail. Section 5 highlights the performance of Pace, and Section 6 showcases important features of Pace, especially use-cases and the benefits of developing in Python. Section 7 documents the limitation of Pace, and finally we summarize in Section 8.

## 2   Python-based DSL: separation of concerns

### 2.1   A modern DSL: requirements

The software productivity gap problem can be described as an issue with the diverse set of skills required by climate modelers to implement a production-grade climate and weather model, from discretizing the underlying physical equations all the way to the optimization details of a given hardware architecture. This process can be roughly split in two:

1. Climate and weather modeling: scientific motivation, algorithmic design, numerical implementation.

2. Performance development: adapting the implementation to a set of given hardware architecture, optimization to reach useful simulation time.

There is overlap between these; some algorithmic designs are better suited to certain hardware architectures, implementation details can be changed to improve model speed, but this breakdown is a useful heuristic for the development of weather and climate models.

The design of a modern domain-specific language needs to respond to both classes of users. For climate modelers a DSL should:

- Be easy to use, complete with debugging tools and simple methods to extracting scientific results.

- Allow easy ways to implement new features and or ways to escape the DSL as new methods are developed.

- Enable quick development round-trip.

- Run with optimal performance.

- Improve development by simplifying the implementation of common code patterns.

For the performance developers it should:

- Leverage a proven host-language to strengthen basic development of a compiler tool-chain and work on solid ground.

- Build a maintainable and extensible set of compiler elements in order to keep up with novel and emerging hardware architectures.

– Ensure the presence of a lower-level interface to perform generic or custom optimizations.

Python stands as a good candidate language to host a DSL given its robustness, ease-of-use, wide adoption across a range of research and industry groups, large ecosystem of pre-existing tools and packages, and capacity for introspection as an interpreted language. Furthermore, in the weather and climate community Python has established itself as the lingua franca for analysis, visualization and post-processing. But Python has serious limitations in terms of execution performance: designed primarily as a scripting language, Python alone cannot achieve the performance required by workloads running on large HPC systems.

The solution is to escape Python at runtime and leverage compiled code. This is a technique already heavily used by frameworks such as Cython (Behnel et al., 2011) and Numba (Lam et al., 2015) (see e.g. Augier et al., 2021). The DSL compiler is responsible for translating and compiling (transpiling) code written in Python into another, more performance-oriented language. The generated code can be tailored and optimized to a specified hardware target via hardware-specific compiler backends.

## 2.2 Related work

For this work we use the GT4Py DSL compiler because the DSL provides a single-source solution for portable performance with the correct abstractions for weather and climate models. There are, however, other approaches using Python at different levels of abstraction that are worth discussing.

Cython (Behnel et al., 2011) is an optimizing static compiler based on Python and the Pyrex language. It is a powerful tool for accelerating Python code, but it lacks the portability and high level abstractions we seek. Other packages provide portability, but still lack the domain-level abstractions to simplify the language for climate scientists. An example of such a package is Numba (Lam et al., 2015), which is a just-in-time (JIT) compiler for Python and Numpy code. JAX (Bradbury et al., 2018) similarly lacks the weather and climate abstractions we desire in the frontend DSL language, but brings with it convenient features such as adjoint capability. Other application-building frameworks such as Exasim (Vila-Pérez et al., 2022), FEniCS (Alnaes et al., 2015), and Dedalus (Burns et al., 2020) mostly operate on the partial differential equation (PDE) level, which allows for more automated model development at the expense of flexibility in the discretization. In order to faithfully reproduce all aspects of the FV3 dynamical core and GFDL cloud microphysics we required greater flexibility than these frameworks provided. Thus, the DSL we are targeting works at a lower level on the mathematical representation after the PDE has been discretized.

## 2.3 GT4Py: a Python-based DSL

Our implementation of the domain-specific language described above is called GridTools for Python, or GT4Py. Developed in partnership with the Swiss National Supercomputing Centre (CSCS), it defines a DSL on top of Python. The code is analyzed and compiled into a C++ or CUDA executable that is bound back to the original Python, creating a seamless experience for the modelers but enabling fast and optimized execution (Fig 1). On top of those performance backends, GT4Py also provides

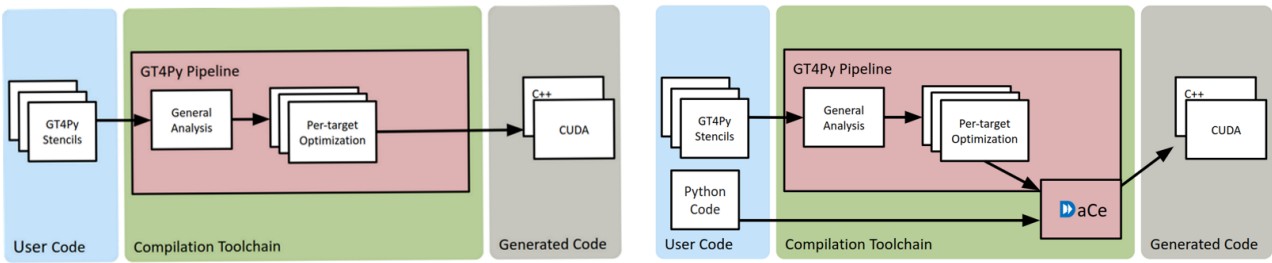

**Figure 1.** Workflow pipeline of GT4Py (left) and GT4Py combined with DaCe (right). User code is analyzed, optimized, and translated into hardware target specific optimized code. In the case of DaCe, the Python code in between GT4Py stencils (control flow) is also included in the translation. This can lead to significant performance improvements.

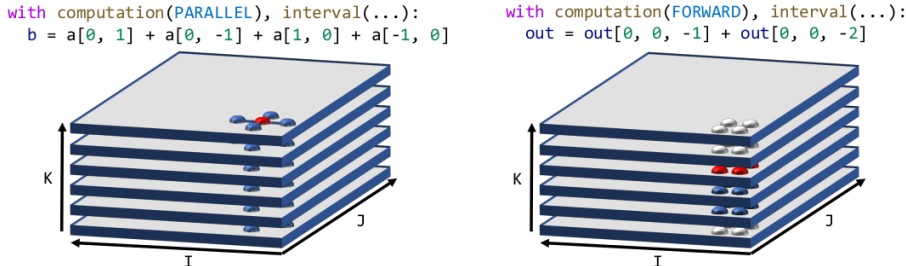

**Figure 2.** Two typical computational patterns in weather and climate models: On the left is a horizontal stencil, on the right is a vertical solver. Figure from Ben-Nun et al. (2022)

a backend to generate Numpy code, which is useful for debugging and quick development as no compilation is required. One GT4Py backend leverages the DaCe (Ben-Nun et al., 2019) framework to be able to also include regular Python code in between DSL code in the translation.

GT4Py operates through the use of stencils: inside a GT4Py stencil each grid cell in an array is modified according to neighboring cells based on a fixed pattern. In numerical modelling of weather and climate two major computational patterns

emerge due to the reliance on three-dimensional structured or unstructured grids: computations with dependencies on the local horizontal neighbourhood of a grid cell and vertical solvers with column dependencies as shown in Figure 2. These two stencil computational patterns therefore form the basis of GT4Py's design.

GT4Py stencils treat the horizontal dimensions differently from the vertical: they always execute in parallel over the entire horizontal domain. In the vertical dimension the order of execution may be specified with the `computation` keyword, and the

vertical range is set with `interval`. A horizontal stencil can be written `with computation(PARALLEL)` to parallelize over the vertical domain as well if there are no vertical loop-carried dependencies. A column-based stencil that calculates up or down the k-dimension, on the other hand, can be implemented sequentially in k `with computation(FORWARD)` or `BACKWARD`, respectively. Each stencil can also have multiple `computation` and `interval` blocks for flow control.

```
 1: def compute_kinetic_energy(
 2:     vc: FloatField,
 3:     uc: FloatField,
 4:     cosa: FloatFieldIJ,
 5:     rsina: FloatFieldIJ,
 6:     v: FloatField,
 7:     vc_contra: FloatField,
 8:     u: FloatField,
 9:     uc_contra: FloatField,
10:     dx: FloatFieldIJ,
11:     dxa: FloatFieldIJ,
12:     rdx: FloatFieldIJ,
13:     dy: FloatFieldIJ,
14:     dya: FloatFieldIJ,
15:     rdy: FloatFieldIJ,
16:     dt_kinetic_energy_on_cell_corners: FloatField,
17:     dt: float,
18: ):
19:     with computation(PARALLEL), interval(...):
20:         ub_contra, vb_contra = interpolate_uc_vc_to_cell_corners(
21:             uc, vc, cosa, rsina, uc_contra, vc_contra
22:         )
23:         advected_v = advect_v_along_y(v, vb_contra, rdy=rdy, dy=dy, dya=dya, dt=dt)
24:         advected_u = advect_u_along_x(u, ub_contra, rdx=rdx, dx=dx, dxa=dxa, dt=dt)
25:         dt_kinetic_energy_on_cell_corners = (
26:             0.5 * dt * (ub_contra * advected_u + vb_contra * advected_v)
27:         )
28:         dt_kinetic_energy_on_cell_corners = all_corners_ke(
29:             dt_kinetic_energy_on_cell_corners, u, v, uc_contra, vc_contra, dt
30:         )
```

.

**Figure 3.** An example stencil definition function that computes the kinetic energy on cell corners. The functions `advect_v_along_y`, `advect_u_along_x`, and `all_corners_ke` are defined outside the stencil and the DSL compiler inlines the relevant code. FloatField and FloatFieldIJ are GT4Py-specific data types for 3d and 2d fields. `FloatField` is a type which is used to declare three-dimensional fields of configurable floating point precision.

Because stencils are applied to each point in a 3D grid, all indexation within a stencil is relative to the "current" computed grid point, i.e. `Array[0,0,0] - Array[1,0,0]` takes each array element and subtracts the array element immediately to its "right" along the x-axis. This method of indexing three-dimensional arrays allows the modelers to use the same indexing convention (I, J, K), irrespective of the actual storage layout in memory.

GT4Py also allows zero-cost function calls, enabling more readable and reusable code within models. Extents of the computation are automatically determined by the DSL, including through these function calls. Figure 3 shows an example stencil function that executes in parallel over the entire vertical dimension. The domain of dependence for intermediate variables (`advected_u` and `advected_v`) is automatically determined by GT4Py. In this example, the advection helper functions take a horizontal difference of the `u` and `v` contravariant velocity components `ub_contra` and `vb_contra`, respectively. As a result, the interpolation function which computes these values is applied on a larger domain than the final operation in the stencil.

In order to implement the FV3 dynamical core we had to extend the stencil concept multiple times, enabling 4- and 2-dimensional arrays, loops over the k-axis, and specialized handling of horizontal subdomains. Because GT4Py is written in Python, those extensions were easy to develop and they will be discussed further in Sections 4 and 5.

GT4Py is able to optimize the performance of stencils, however some Python overhead remains in the code linking the stencil operations together. In order to resolve this and optimize the entire Pace model we have incorporated DaCe (Ben-Nun et al., 2019) as a backend for Pace. Using this framework both GT4Py stencils and raw Python code are exposed to optimization and transpilation to high-performance C++ or CUDA executables, removing all Python overhead seamlessly for the modelers. Section 5 contains more information on code optimizations, while Ben-Nun et al. (2022) details the performance optimizations more thoroughly.

## 3 Pace

Pace is a GT4Py implementation of the nonhydrostatic FV3 dynamical core and GFDL microphysics (Putman and Lin, 2007; Harris et al., 2021; Chen and Lin, 2013; Zhou et al., 2019). It is based on the same version of the National Oceanic and Atmospheric Administration (NOAA) Unified Forecast System (UFS) model as McGibbon et al. (2021), forked from the UFS respository v1 in December 2019 (Zhou et al., 2019), and is nearly identical to the dynamical core used in SHiELD (Harris et al., 2020b). At present Pace only supports nonhydrostatic, uniform-resolution simulations, with a restricted set of subgrid reconstruction schemes (`hord` and `kord` values). Pace has the ability to read initial conditions generated by the Fortran model and other saved outputs, and can also generate initial conditions for analytic test cases. Currently Pace supports 6-tile gnomonic cubed-sphere grids and single-tile orthogonal, doubly-periodic grids, though only at uniform resolution. Future development will enable nested and stretched grids as described in Harris and Lin (2013) and will integrate the rest of the physics parameterizations (Zhou et al., 2019). Pace is MPI-enabled, allowing it to run in parallel, but can also run using a "serial" communicator, running each rank in serial and saving data files to mock MPI communication.

### 3.1 A Modular Model

Pace is designed to be modular; each model component of Pace (e.g. dynamics, microphysics, utilities, DSL integration) exists as a separate package. Computationally-focused packages like the dynamics contain heirarchies of component modules (Figure 4). These components provide clear boundaries to document and change model behavior. For example, the horizontal transport scheme used by FV3 (Putman and Lin, 2007; Lin and Rood, 1996) takes any one-dimensional finite volume subgrid reconstruction scheme satisfying certain numerical conditions, and can extend it to two dimensions. Within Pace, one-dimensional subgrid reconstruction code is contained in the XPPM and YPPM modules (X- and Y-Piecewise Parabolic Methods, respectively). These modules take in scalar gridcell-mean values and Courant numbers (speed as a fraction of gridcell width) defined on transport interfaces, and return the average value of the scalar within the section of gridcell to be advected through the cell interface. The FiniteVolumeTransport class extends these one-dimensional subgrid reconstruction schemes to produce two-dimensional horizontal fluxes. This allows a scientist to modify the behavior of the dynamics by replacing only the XPPM and

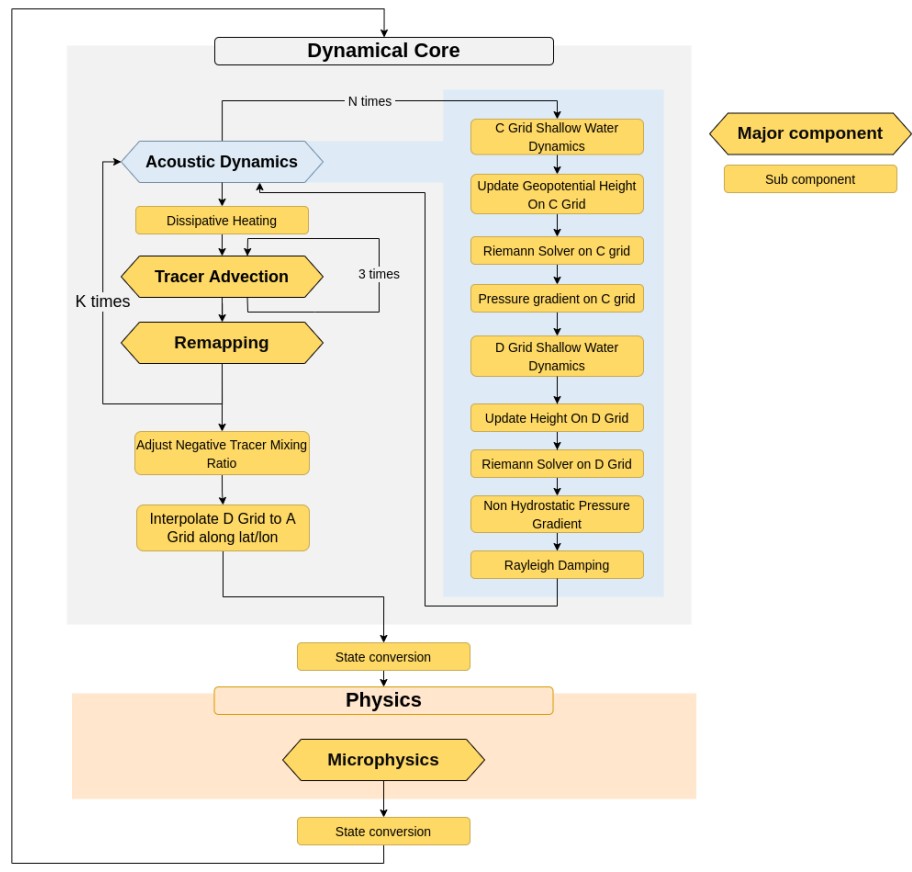

**Figure 4.** Internal structure of the Pace dynamical core. To reduce instabilities from the lagrangian vertical coordinate the acoustic dynamics, tracer advection, and vertical remapping are sub-stepped, executing K times during a model timestep. The acoustic dynamics are additionally sub-stepped to ensure numerical stability with respect to sound waves.

YPPM components, for example with a cubic reconstruction scheme or a subgrid reconstruction scheme based on machine learning.

To maintain simplicity of the code and facilitate the separation of compile- and run-time, Pace uses a simple object-oriented framework that expresses nearly all of the internal computational components as initializable functions, as shown in Figure 5. Each of these functions is defined as a Python class with an initialization method and a call method. Initialization performs necessary setup including memory allocation and compilation of DSL code. Each component's initialization code defines the component's temporary variables, compiles or loads its stencils, and recursively initializes sub-components. Once initialized, the component may be called the same as any other Python function. This is similar to Fortran model structures where computational code is paired with initialize and finalize subroutines.

Stencils are compiled when components are initialized, using only explicitly-passed configuration data. Pace uses the factory pattern (through the `stencil_factory` argument) to reduce compilation-specific logic as much as possible within compu-

```
 1: class ComputeKineticEnergy:
 2:     def __init__(self,
 3:         stencil_factory: StencilFactory,
 4:         grid_data: GridData,
 5:         config: DGridShallowWaterLagrangianDynamicsConfig,
 6:     ):
 7:         self.grid_data = grid_data
 8:         self._compute_kinetic_energy = stencil_factory.from_dims_halo(
 9:             func=compute_kinetic_energy,
10:             compute_dims=[X_INTERFACE_DIM, Y_INTERFACE_DIM, Z_DIM],
11:             externals={
12:                 "iord": config.hord_mt,
13:                 "jord": config.hord_mt,
14:                 "mord": config.hord_mt,
15:                 "xt_minmax": False,
16:                 "yt_minmax": False,
17:             },
18:         )
19:
20:     def __call__(self, vc, uc, v, vc_contra, u, uc_contra,
     dt_kinetic_energy_on_cell_corners, dt):
21:         self._compute_kinetic_energy(
22:             vc=vc,
23:             uc=uc,
24:             cosa=self.grid_data.cosa,
25:             rsina=self.grid_data.rsina,
26:             v=v,
27:             vc_contra=vc_contra,
28:             u=u,
29:             uc_contra=uc_contra,
30:             dx=self.grid_data.dx,
31:             dxa=self.grid_data.dxa,
32:             rdx=self.grid_data.rdx,
33:             dy=self.grid_data.dy,
34:             dya=self.grid_data.dya,
35:             rdy=self.grid_data.rdy,
36:             dt_kinetic_energy_on_cell_corners=dt_kinetic_energy_on_cell_corners,
37:             dt=dt,
38:         )
```

.

**Figure 5.** A class that compiles and runs the stencil defined in Figure 3. The `__init__` method initializes the `compute_kinetic_energy` stencil using the stencil definition function defined in Figure 3, an output domain, and the constants used by the stencil, and the `__call__` method executes the stencil when the resulting object is invoked at runtime.

tational components. The stencil factory class implements the code responsible for allocating and compiling stencils, allowing model code to instead focus on computational motifs. In lines 8-18 of Figure 5 we can see this factory at work compiling the

`compute_kinetic_energy` stencil shown in Figure 3. The `from_dims_halo` method takes a stencil function and a set of dimensions (either cell-centers or cell-interfaces) to execute over, and returns a compiled stencil that writes its outputs over the compute domain, in this case on cell corners (x and y interfaces). Output can be extended into the computational halos with an optional `compute_halos` argument indicating how many halo points to write (not shown). We also see how compile-time constants are passed as `externals` to the compilation method. These constants can be set by the model configuration,

extracted from the domain decomposition, or passed as arguments to the initialization method, and have the benefit of being

treated as compile-time constants. Because configuration settings such as domain-decomposition and namelist settings are now known at compile-time they also provide further avenues for the DSL backend to optimize performance.

The factory pattern used here is incredibly powerful when debugging the model. For example, if a model developer finds that at some point a variable `air_pressure` used by many routines has gone negative in the model, that developer can temporarily insert code to immediately raise an exception if *any* stencil writes a negative value to a variable named `air_pressure`. Or that model developer could alter every stencil to write its inputs and outputs to a netCDF file when executed, while only having to modify the code used for the stencil factory.

## 3.2 Powerful Testing

The introspection power of Python is used to great effect in the testing code for Pace. For example, we would like to ensure that Python array allocation only happens when initializing our model, and not at all when it is called. To do this, we write a test which initializes the model, then replaces the storage allocation routines of GT4Py with routines which raise an exception if called before finally calling the model. If any arrays were allocated at call time, an exception would be raised and the test would fail.

We also want to ensure that the Pace model components are not stateful. The dynamical core, for example, has many temporary storage arrays assigned to it whose initial value should not matter when calling the model. However, a user could easily fail to initialize an array with zeros when they mean to, causing a bug in the model. We have a test which calls the dynamical core with the same state either once or twice, and compares the value of all temporary data in the model between these two cases. It does this by dynamically crawling the Python object structure and comparing all array data. If any data differs between those two cases, it not only tells us that there is a bug but also exactly which temporary has a bug - the one which is accessed first out of the ones which differ. Without such a test, it could take days, weeks, or months for a scientist to find the source of such an error, assuming they notice the presence of the bug.

## 4 Porting the Model

Atmospheric models are large computational codes, making it difficult to determine the source of a bug given errors in model outputs. In order to port FV3 and the physics parameterizations we first segment the Fortran code into smaller units of code which can be ported and tested independently. Typically each unit encompasses a particular Fortran subroutine, but larger or more complicated subroutines may be broken down further to ease validation. We use the Serialbox library to extract the inputs and outputs from each of these Fortran units. We place Serialbox compiler directives before each unit of code to serialize the inputs to that unit, and similarly insert directives after each unit to extract its outputs. For a given test case we can then run the Fortran model to generate test data for that case and model configuration.

GT4Py stencils calculate over 3D volumes, so in our porting process we initially wrote each individual stencil to replicate a Fortran do-loop over the i, j, and k spatial dimensions. A ported unit of code may use multiple stencils if there are computa-

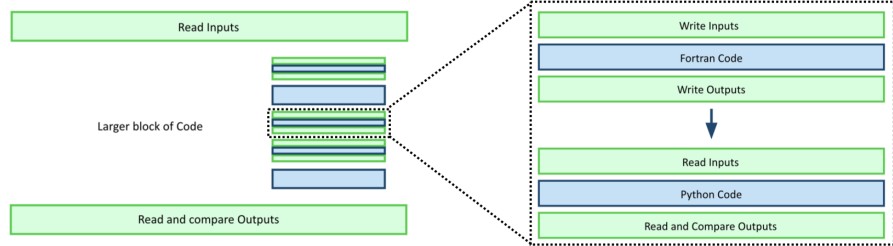

**Figure 6.** Schematic of our porting strategy: We port small units of code, test that they validate against the original Fortran, and then assemble them into larger model components we then validate, building up to a full model port.

tions over different horizontal domains, or to increase readability of the code. In order to minimize GPU kernel launches we subsequently merged stencils that executed over the same spatial extents where possible.

### 4.1 Extending the DSL

In the process of writing Pace we needed to extend GT4Py in order to express the Fortran code in the DSL. FV3 discretizes the Earth into a gnomonic cubed-sphere (Putman and Lin, 2007). GT4Py could only apply the same operations uniformly across the horizontal domain of a stencil. We added the ability to execute code on stencil subdomains to the GT4Py DSL language in order to perform special handling required near the corners and edges of the cubed-sphere tiles. Specifically, we included `horizontal` and `region` as stencil control keywords (similar to `computation` and `interval`) as shown in Figure 7.

`with horizontal` specifies a horizontally restricted block of code, and `region` specifies the extent of that subdomain through Numpy-like array slicing in the I and J dimensions. Though it does violate the GridTools concept of a stencil, this allows us to fully port the dynamical core and combine stencils to reproduce a natural amount of Fortran code. This also has the benefit of increasing the readability of the code: in Fortran the corner and edge computations are handled by the common pattern of if-conditionals inside of nested for-loops, while in Pace the horizontal regions are used almost exclusively for this

purpose.

Another example is in the vertical remapping component of the model. The FV3 dynamical core uses a Lagrangian vertical coordinate which is regularly remapped to its original Eulerian grid using the piecewise parabolic method (Lin, 2004). The remapping process requires a double k-loop over the vertical dimension: an inner loop over the deformed vertical levels to sum their contributions to an Eulerian level, and an outer loop over those Eulerian levels. When we initially ported the remapping

code GT4Py did not support `for` or `while` loops inside of stencils, so we wrote the inner loop as a stencil over the deformed k-levels and the outer loop in plain Python. While this implementation worked algorithmically, calling a stencil inside of a for-loop has a large penalty in model speed due to the repeated kernel launches and the removal of the loop structure from the DSL optimization path. To resolve this we added while-loops to GT4Py stencils. This allows us to consolidate the remapping code into one stencil over the Eulerian k-levels, remove the Python for-loop, and expose the entire remapping scheme to the

DSL compiler. This reduced the run-time of the remapping step by over an order of magnitude.

```
1: @gtscript.stencil(...)
2: def set_field(field: Field[float]):
3:     with computation(PARALLEL), interval(...):
4:         field = 0.0
5:         with horizontal(region[i_start, :]):
6:             field = 1.0
7:         with horizontal(region[i_start + 1, :]):
8:             field = 2.0
9:         with horizontal(region[i_start + 2, :]):
10:            field = 3.0
```

.

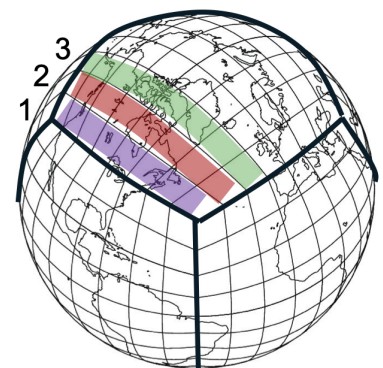

**Figure 7.** Example illustration of how horizontal regions in GT4Py can be used to specilize computations in certain sections of the computational domain: The first region assigns 1.0 to the first row of the array, the second region assigns 2.0 to the second row, and the third assigns 3.0 to the third row.

GTPy is able to cover a large number of algorithmic motifs used in weather and climate models. This section illustrated two examples where the DSL has been extended in order to express motifs which were present in FV3GFS. There is also an on-going effort to extend GT4Py to unstructured grid computations. Nevertheless, any DSL will naturally be restricted from covering some algorithmic motifs. For GT4Py these include reductions (e.g. computing horiztonal integrals over a field), interpolation (e.g. for multi-grid), and search patterns (e.g. looking for a neighboring grid cell with certain properties). Whether these patterns should fall into the scope of the GT4Py DSL or another framework is a design decision, which can be weighed for example against compiling these patterns with DaCe.

### 4.2 Model Validation

Each unit of code is validated by running it with the input data serialized from runs of the Fortran model, and comparing the outputs of our code to the serialized Fortran outputs. If a ported unit is particularly large (e.g. vertical remapping, grid generation) we serialize data from components of the larger unit and test the equivalent Pace components to reduce the complexity of each test. We use two sets of initial atmospheric conditions as our test cases, each with a corresponding configuration namelist. Our "standard" test case is generated from NCEP reanalysis data from 0Z on August 1, 2016 (as in McGibbon et al., 2021),

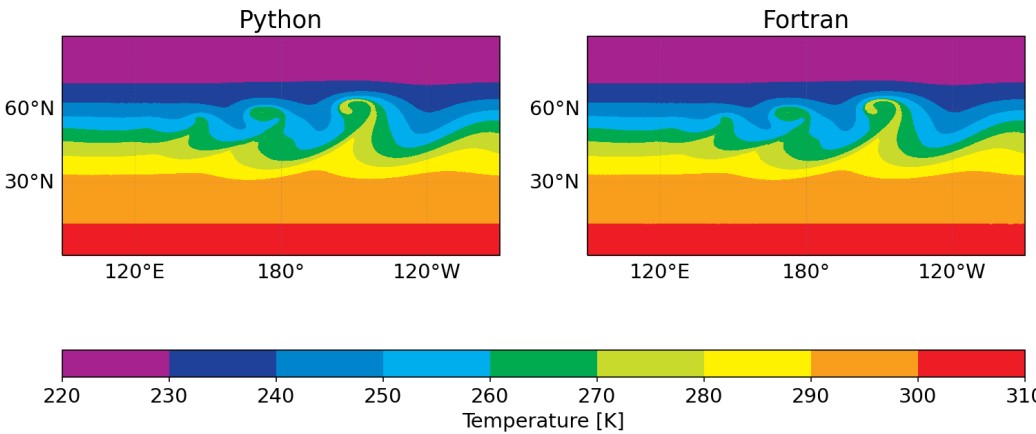

**Figure 8.** Detail of 850 mb temperature of the baroclinic instability simulated with the Pace (left) and Fortran (right) dynamical cores on day 9 at 10-km resolution. These results can be compared to Fig. 6 of Jablonowski and Williamson (2006) and show how well our model replicates the original Fortran.

and the other is the baroclinic instability test case described in Jablonowski and Williamson (2006). Initial versions of the DSL

code were tested on the standard case run on 6 MPI ranks (one per tile) and a 12 by 12 horizontal grid on each tile face with 79 vertical levels. We also test with a 54-rank domain decomposition, as this gives each tile face a rank for each corner and edge as well as a central rank with no edges. Our tests thus cover the range of edge and corner handling required of a given rank from no corners or edges to all eight corners and edges.

In many cases the Pace outputs can be brought near roundoff error to the Fortran outputs. However, there are some instances

where changes to the order of operations or the use of transcendental functions makes such reproducibility impossible, so we must choose a validation threshold for our code. We do this by perturbing the inputs to the Fortran code by small, floating-point differences, and comparing the outputs. This leads us to adopt relative differences of $10^{-14}$ as our default validation threshold. If we find validation errors greater than our default we test the components of the offending code to see what operations introduce discrepancies between Pace and Fortran. Often these differences are due to errors introduced during porting and

we can reconcile Pace with the Fortran code. If these differences are instead due to algorithmic changes (reordering, etc.) we set the error tolerance for that piece of code based on the underlying Fortran/Python difference. For example, in the vertical remapping scheme FV3 makes use of multiple `goto` statements for its control flow. These are not available in GT4Py, and so some code has been rearranged to replicate the original algorithm, introducing small deviations from the Fortran outputs.

When we are confident that each component of the model accurately reproduces the Fortran version, we test larger combi-

275 nations of these units to ensure the implementation of these larger modules also matches Fortran, as illustrated in Figure 6. For example, after validating the cubic-spline vertical interpolation code and the code that calculates the contributions from deformed, Lagrangian pressure levels to the remapped Eulerian levels, we then validate the code to vertically remap a single variable from Lagrangian to Eulerian pressure levels. When this code validates, along with the tracer remapping, saturation

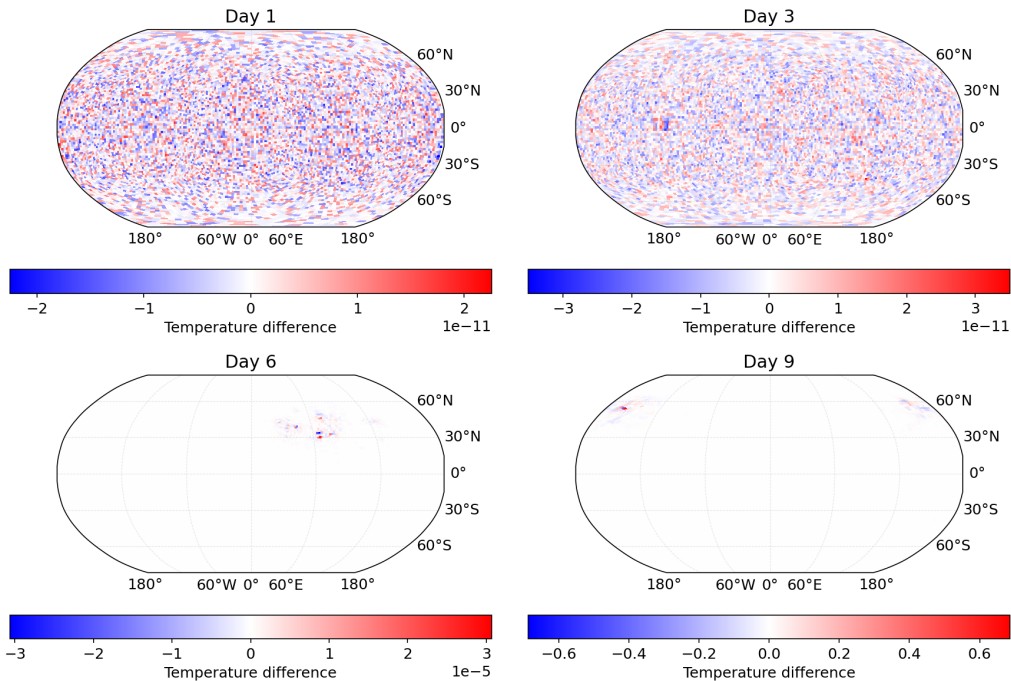

**Figure 9.** 850 mb temperature difference between Pace and baseline Fortran simulations of a baroclinic instability at 200 km resolution after 1, 3, 6, and 9 model days, showing a good match between the two models.

adjustment, and moist potential temperature adjustment code, we can then validate the entire vertical remapping scheme. In this way we hierarchically assemble and validate Pace against the Fortran code. These validation tests are incorporated into our suite of continuous integration tests to ensure that future developments and code changes do not affect model validity.

Once the full Pace dynamical core passes these unit tests we run the Jablonowski and Williamson (2006) baroclinic instability test case for 9 model days and compare the results against the same test run in Fortran. In this test case, the dynamical core is initialized with zonally symmetric steady-state winds, on top of which a Gaussian perturbation of the zonal wind is superimposed in the northern mid-latitudes, triggering the evolution of a baroclinic wave. In Figure 8 we show the region of interest in a 10-km resolution simulation (960x960x80 grid cells per tile, run on 150 MPI ranks) of the same instability from Pace and the reference Fortran model. We see that the model behavior is consistent with the reference Fortran model and replicates the results from the highest horizontal resolution in Jablonowski and Williamson (2006) well. Figure 9 shows the difference in 850 mb temperature between Pace and Fortran running the test case at 200-km (48x48x80 grid cells per tile) resolution. Both models were run with 6 MPI ranks. The Pace dycore matches the Fortran code closely early on, with random errors on the order of $10^{-11}$ after three model days of integration. Due to the nature of the baroclinic instability these small errors do eventually grow to the order of $10^{-1}$ after 9 simulation days (3888 timesteps), but even on day 6 the errors are only on the order of $10^{-5}$. Based on these results we are confident that Pace accurately reproduces the Fortran model code.

| Tile Size | Scaling | Fortran | | Pace CPU | | | Pace GPU | | |
|---|---|---|---|---|---|---|---|---|---|
| | | Time [s] | Scaling | Time [s] | Scaling | Speedup | Time [s] | Scaling | Speedup |
| 108x108x80 | – | 3.58 | – | 16.00 | – | 0.22 | 1.98 | – | 1.81 |
| 128x128x80 | x1.40 | 4.66 | 1.30 | 22.25 | 1.39 | 0.21 | 2.34 | 1.18 | 1.99 |
| 192x192x80 | x2.25 | 12.74 | 3.56 | 48.07 | 3.00 | 0.27 | 3.98 | 2.01 | **3.20** |

**Table 1.** Performance metrics comparing the Pace dynamical core with the Fortran reference code. The size (in number of gridpoints) of each face (tile) of the cubed sphere grid is increased from row to row. Each of the configurations is run on 6 compute nodes, one compute node per face of the cubed-sphere grid. Essentially this corresponds to global simulations of decreasing grid spacing of 96 km, 72 km, and 48 km, respectively. The time measurements are the execution time of one invocation of the dynamical core (see Fig. 4).

Pace has only been validated using double precision floating point arithmetic and values. All performance results shown in the next section have been measured using double precision.

## 5 Model Performance

All experiments were conducted on the Piz Daint supercomputer at CSCS. Piz Daint contains 5704 Cray XC50 nodes, with an Intel Xeon E5-2690 v3 12-core CPU, one NVIDIA Tesla P100 GPU (16 GB RAM), and 64 GB of host RAM on each node. The nodes are connected via the Cray Aries interconnect. We are using Python 3.8.2 with Pace revision 0.2. the generated code was compiled with CUDA 11.2 and GCC 9.3.0. To expose the full node architecture to the DSL optimization scheme we run Pace with one MPI rank per node for both CPU and GPU backends. The reference Fortran model was compiled with Intel IFORT version 19.1.3.304. We run the Fortran code in its optimal configuration of 6 ranks per node and 4 threads per rank, fully utilizing all 24 virtual cores available with hyperthreading.

Table 1 shows our model performance and strong scaling results, comparing the absolute runtime of the Pace dynamical core on both CPU and GPU architectures with the original Fortran implementation. Figure 11 presents these data graphically. We show our weak scaling results in Figure 10. The microphysics implementation in Pace takes a negligible amount of time ($\lesssim 1$ %), so we focus our discussion on the dynamical core.

Our weak-scaling results (Figure 10) demonstrate a speed up of 3.6x against Fortran that is nearly constant across simulation scales. The slight decrease in runtime at higher scales is due to the heavier computational load on the limited case of 6 MPI-ranks; when running on 6 MPI ranks (one rank per tile face) FV3's corner and edge handling has to be computed on every rank, while at higher scales that computational load is better spread across MPI ranks. Section 5.3 discusses this in more detail. Overall we see that Pace exhibits perfect weak scaling, which validates the capacity of a GPU-running model to simulate at km-scale resolutions in an efficient way.

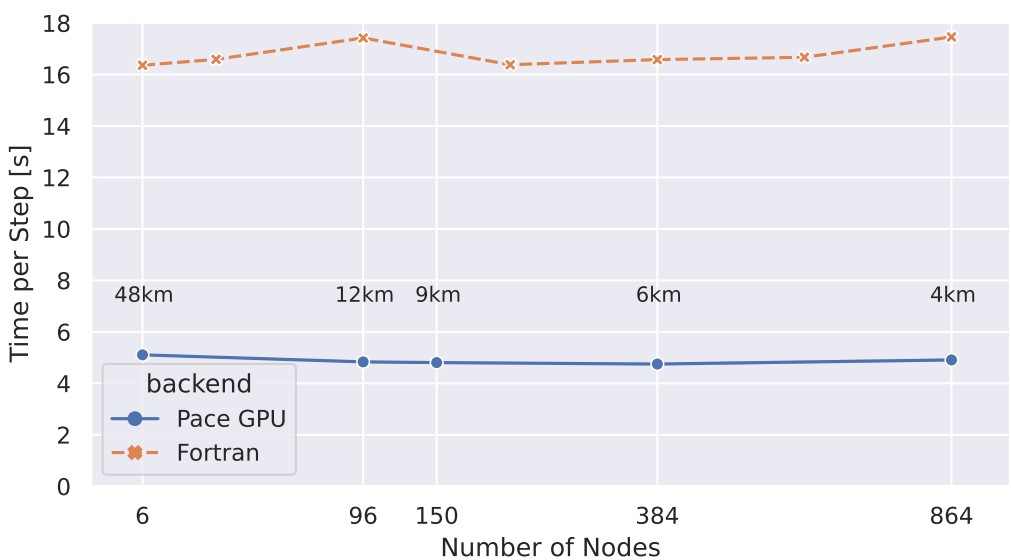

**Figure 10.** Weak scaling analysis from Ben-Nun et al. (2022): Execution time for one invocation of the dynamical core (see Fig. 4) is shown as a function of an increasing number of compute nodes (each with one GPU). In this setup Pace is run with one MPI rank per node, while the Fortran code is run with 6 MPI ranks per node and 4 threads per rank. In this weak scaling experiment, the number of grid points per GPU are kept constant at 192x192x79. As more nodes are added, the resolution of these global simulations increases. The average grid spacing is indicated in text and a maximum of 4 km is achieved when running on 864 nodes.

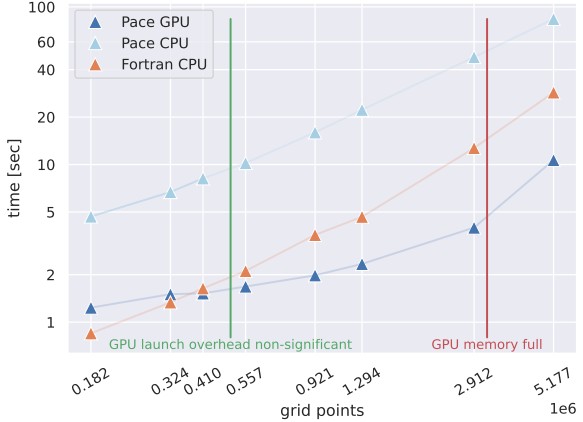

**Figure 11.** Dynamical core runtime for varying domain sizes, expanded from Table 1. The time measurements indicate the execution time of one invocation of the dynamical core (see Fig. 4)

Since we observe perfect weak-scalability, meaning an increase in compute nodes working on a problem does not affect the total time we can determine that the total time only scales with the work per node. Since the work is increasing linearly with the number of gridpoints per node we show a detailed analysis of the scalability of pace with varying work per node.

The domain size scaling experiments shown in Figure 11 show increasing performance gains on GPU as larger domains are simulated on each compute node. This is expected, as GPUs are well fitted to high domain sizes since they are capable of higher throughput than CPUs when optimized accordingly. As is well known and described for example in Fuhrer et al. (2018), offloading computation to GPUs comes with a non-insignificant startup cost that only starts to pay off if enough work is done on accelerators. Pace is no exception to this rule. We can see a regime where not enough work is done to justify the startup cost.

A detailed view of the scaling numbers for the most relevant region is summarized in Table 1. When we maximize the amount of data on each node - shortly before we run out of memory on the GPU - Pace achieves a speed up of 3.2x in comparison to Fortran. The CPU version, on the other hand, is 0.2x the speed of the Fortran code. This is due to a combination of two factors:

- the Fortran model we ported is itself highly-optimized for CPU,

- the optimization effort was geared toward demonstrating the viability of the Python-DSL for GPU usage.

Despite focusing on GPU optimization, some of the optimization methods also improved CPU performance, and the functional CPU backend does demonstrate portability. While outside the scope of this paper, work to improve CPU performance is now underway.

Model performance is a core motivation for GPU acceleration and for adopting a DSL, and it is encouraging to see a significant speedup between Pace-GPU and Fortran versions. While the CPU performance has not been our focus to date, it is the next priority for Pace development. We discuss our optimization strategies and explain how model performance influenced our decision process in porting the model in the following subsections. More detailed performance results and a thorough analysis are available in Ben-Nun et al. (2022). The version of the model code and supporting framework for Ben-Nun et al. (2022) is slightly earlier, hence a slight difference in absolute numbers, but the methodology remains the same.

### 5.1 Optimizations

Because GT4Py has multiple backends for various target architectures there is no one performance number that captures the entirety of our approach. Nonetheless, the aggregate performance across backends indicates the power of the DSL paradigm through performance portability. With the capability of code-generating for specific hardware targets, we are not limited to single, catch-all solutions in our user-facing code. Instead we can generate optimized code for each type of hardware individually through backend logic, allowing user code to focus on numerical details. As an example, on CPU architectures it usually pays off to use coarse-grained parallelism such that threads are assigned to different subdomains, while on GPUs the parallelization strategy involves having blocks of threads execute subdomains.

Weather and climate models are written with large configuration files, namelists, to allow for flexible use. In order to support all the possible configurations, standard models use many conditionals that can not be resolved at compile time, such as

which subgrid reconstruction to use for tracer advection. This limits how aggressively the low-level compiler can optimize the model. Code-generation from a DSL allows us to code-generate and compile the model for a specific namelist configuration to circumvent this problem.

Furthermore the DSL compiler is able to apply domain-specific optimizations under certain conditions on a per stencil level before generating code at all. These include:

1. *Reduce main memory accesses*: Replace 3D temporary fields used for intermediate results that are stored in global memory with smaller buffers that allow for more reuse and faster access.

2. *Inlining*: Fully inline both function calls as well as nested conditionals, replacing function calls and branching conditionals with the relevant code at compile time and removing the performance overhead of these code patterns.

3. *Pruning*: Analyze the code based on all the compile-time constants provided and prune unreachable branches of code.

4. *Fusion*: Fuse numerical operators defined in separate functions and contexts into single kernel calls, as long as that does not create race conditions. This optimizes performance by reducing the amount of synchronization needed as well as giving the underlying general purpose compiler more flexibility.

## 5.2    DaCe

In order to achieve good performance in a Python driven environment it is crucial to minimize the overhead moving from the driver language to the compiled language executable. Since our implementation of the model is a series of compiled stencils called from Python, the performance overhead linked to calling the compiled stencils from Python scales with the number of stencils called. A secondary issue is that with the fragmentation into individual stencils we limit the optimization potential of the DSL, as it is limited to the scope of a single stencil for the optimizations described above.

We leverage DaCe (Ben-Nun et al., 2019) to address those shortcomings, providing a full-program optimization framework. With our DaCe backend in GT4Py we are able to compile the entire loop over timesteps into a single executable called from Python, thus completely removing all Python to C call overhead during our simulation. With DaCe as our backend we are able to leverage custom optimization for our code including improving the computational layout, improving where and how memory allocation happens, scheduling computation to improve parallelism as well as increasing data locality and improving the pressure on global memory. A detailed explanation of our approach and how it affected performance can be found in Ben-Nun et al. (2022).

## 5.3    Corner and edge handling

As discussed in Section 4.1, the cubed-sphere discretization of FV3 (Putman and Lin, 2007) requires special finite difference stencils applied near edges and corners to account for the grid geometry. We added the ability to execute stencils on horizontal subdomains to enable these motifs in GT4Py, which unfortunately has negative performance implications for Pace.

FV3 parallelized the vertical plane K instead of the horizontal IJ ones. Flipping this order of parallelization increases the parallelism many-fold, which is necessary for high throughput on GPUs and other massively parallel architectures. FV3 parallelizes in this way because it is advantageous for their target architectures which are CPUs with branch prediction and large caches. On CPU architectures instruction divergence inside the tight loops due to FV3's corner and edge handling is not an issue, but there is a limit to what can be done in a single-instruction multiple data (SIMD) GPU kernel. Because of their data dependencies these corner and edge operations cannot be executed in parallel with the rest of the code, requiring a separate kernel launch for each of them. This breaks what could be one large and optimized stencil into separate stencils separated by such corner calculations, though there are certain conditions under which we can fuse these operations into the same kernel.

This example shows how algorithmic choices have large impacts on performance. The choice of specialized corner and edge handling made for FV3 fits a CPU architecture well but is sub-optimal for GPU architectures. Changes to the algorithm can be made to alleviate this problem, such as a duo-grid approach demonstrated in Chen (2021). Since Pace is intended to be a one to one port of FV3 in it's current state this is not a development we are focused on at the moment, though it could be a fruitful direction for other DSL models.

## 6  Pace in Action

### 6.1  Driving the model

The driver code in Figure 12 showcases the power of Python as an API language for configuring, compiling, and running computational codes. The `command_line` entry function reads all information needed to run the model from disk, and directly defines the behavior of the Driver class. Within the Driver (not shown), the dynamical core and physics components only have access to the configuration and variables they use, which makes it much easier for a new developer of the model to understand what the code does. This is true all the way down through its components, as described in Section 3.1.

This example also showcases the advantage of working in a widely-used language with a vibrant open source package ecosystem. Python has powerful tools available for defining command-line interfaces. Taking a yaml configuration file and mapping it onto a nested configuration class is as simple as using pyyaml and dacite, which provide this functionality. Python's built-in datetime and timedelta types make it easy to manage model execution time, and external packages such as cftime provide support for a wide range of calendars. Diagnostics storage makes use of the xarray and zarr packages to greatly simplify the code we need to write. This well-established ecosystem of tools maintained by scientists and engineers from a range of disciplines and sectors is incredibly helpful when developing an atmospheric model.

And in turn, having a model written in Python means the tools we have written can be used by anyone who uses the language. This is particularly important given the popularity of Python as a language for scientific analysis. When processing the output of Pace, a scientist has direct access to the tools and numerical code used by the model itself.

```
1:  @click.command()
2:  @click.argument(
3:      "CONFIG_PATH",
4:      required=True,
5:      type=click.Path(exists=True, readable=True, dir_okay=False, resolve_path=True),
6:  )
7:  @click.option(
8:      "--log-rank",
9:      type=click.INT,
10:     help="rank to log from, or all ranks by default, ignored if running without MPI",
11: )
12: @click.option(
13:     "--log-level",
14:     default="info",
15:     help="one of 'debug', 'info', 'warning', 'error', 'critical'",
16: )
17: def command_line(config_path: str, log_rank: Optional[int], log_level: str):
18:     """
19:     Run the driver.
20:
21:     CONFIG_PATH is the path to a DriverConfig yaml file.
22:     """
23:     configure_logging(log_rank=log_rank, log_level=log_level)
24:     logger.info("loading DriverConfig from yaml")
25:     with open(config_path, "r") as f:
26:         config = yaml.safe_load(f)
27:         driver_config = DriverConfig.from_dict(config)
28:     logging.info(f"DriverConfig loaded: {yaml.dump(dataclasses.asdict(driver_config))}")
29:     main(driver_config=driver_config)
```
.

**Figure 12.** The driver function to run Pace from the command line. The `command_line` function is able to gather the configuration and set logging behavior and run the model.

## 6.2 Use Cases

One of the advantages of Python is the ecosystem surrounding it. For example, packages such as NumPy and SciPy make it easy to perform common mathematical and numerical operations and manipulate data; Matplotlib enables interactive visualization
and creates static or animated images; Jupyter Notebook allows users to create and share their computational documents; TensorFlow provides end-to-end libraries for machine learning and artificial intelligence applications. With Pace, we can leverage all these tools to make running, processing, and visualizing climate model output all in one Python script.

Pace also enables novel workflows through the use of Jupyter notebooks. The Pace repository contains an example notebook running one component of Pace. We first initialize a cubed-sphere grid and an idealized atmospheric state, then run the tracer
advection operation and visualize the results. In this case the analytic zonal wind profile should advect the tracer mass once around the Earth in twelve model days, and any deviation from this indicates a problem with the advection code. This workflow is meant to mimic how a model developer could develop, implement, and debug a new advection scheme. We take advantage of the fact that each component in Pace is modular and skip over running the entire model, which enables rapid prototyping without leaving the Python ecosystem. Beyond development, this capability is useful for teaching, allowing students to inspect
elements of the model individually.

```
1:  class Physics:
2:      ...
3:
4:      # GT4Py stencil-based physics:
5:      prepare_microphysics(physics_state)
6:      microph_state = physics_state.microphysics
7:      microphysics(microph_state)
8:      ...
9:
10:     # Machine-learning based microphysics
11:     emulation_model = tf.keras.models.load_model("microphysics.tf")
12:
13:     emulation_dict = prepare_emulation_data(physics_state.microphysics)
14:     predictions = emulation_model(emulation_dict)
15:     model_outputs = unpack_predictions(predictions, emulation_model.output_names, _)
16:     update_physics(physics_state, model_outputs)
```
.

**Figure 13.** Pseudo code outlining how a machine-learned microphysics emulation scheme could be incorporated as a model component in Pace.

## Liquid water column sum

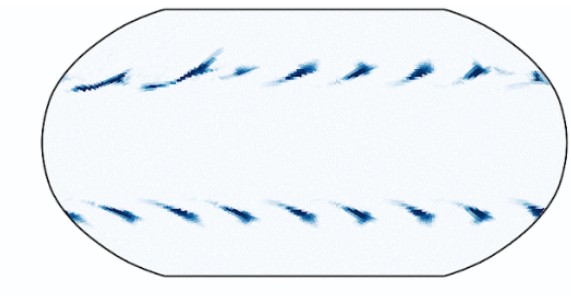

**Figure 14.** Microphysics emulation results: liquid water column sum of a baroclinic instability simulation on day 10 at 200 km resolution

Another benefit of Pace is that it facilitates incorporating novel applications and approaches into the model, such as machine learning emulation. Oftentimes machine learning in climate models requires complicated workflows such as calling Fortran from Python (McGibbon et al., 2021) or calling Python from Fortran. Because Pace is written in Python, we can train and execute an ML model directly, bypassing difficult infrastructure to pass variables between languages. In Figure 13 we show

Pace's ability to replace a stencil-based microphysics scheme with a pre-trained TensorFlow based microphysics emulator. Taking advantage of the modularity and separation of initialization and call time of the individual module, we simply load the TensorFlow model during the initialization of the Physics class and replace the call signature of microphysics. Figure 13 shows an example of the emulator applied to the baroclinic instability test case.

## 7 Limitations

Our research has focused on porting a subset of NOAA's FV3GFS model so as to limit the porting effort and focus on framework development. In particular, the nesting capabilities of FV3 were not ported. Likewise, we forked the model code early in the project and did not pull any updates added to the original Fortran code by NOAA's team. Despite this limiting choice, we believe the two namelist configurations we ported cover a wide range of applications to show that any model code can be effectively ported.

While this paper presents our port of the dynamical core and microphysics, we are actively working on integrating and validating gt4py ports of the full GFS physics suite. Our results integrating the GFDL microphysics scheme (Chen and Lin, 2013) into Pace shows not only show validation and competitive performance, but also demonstrates the feasibility of implementing physics parameterizations using the same strategy deployed for the dynamical core. We have written GT4Py implementations of the remaining physics schemes (PBL, turbulence, shallow convection, sea-ice, land surface, and radiation), and work is 440 ongoing to optimize and integrate them into the Pace model.

Only the baroclinic wave test case was tested at the larger 150-rank configuration described in Section 4.2. A subset of an earlier version of the code was run on 2400 GPU nodes (Ben-Nun et al., 2022) before achieving full validation, alleviating concerns about distributed performance.

Lastly, as explained in Section 5, we have only focused on GPU performance optimization thus far, leaving CPU performance 445 sub-optimal. We are confident that CPU optimization is achievable within the boundary of the current framework and only requires careful engineering.

## 8 Conclusions

We have presented Pace, an open-source performance-portable implementation of the FV3 dynamical core and GFDL cloud microphysics written in Python using the GT4Py domain-specific language. The DSL implementation allows Pace to run on 450 both CPU and GPU architectures, where it achieves high performance. The use of Python as a front-end language lets us write Pace in a modular, productive style and gives us access to Python's powerful testing tools, which allows us to debug and validate Pace.

We have demonstrated our method of porting code from Fortran to Python, building Pace hierarchically from small units of ported code that validate regularly against their Fortran equivalents. Fully porting Pace was a combination of porting the code 455 to the DSL and extending the DSL to cover new algorithmic motifs required by the model, such as while-loops inside stencils and allowing computations to occur on horizontal subdomains. Our testing strategy ensures that our code remains equivalent to the Fortran model throughout our development, including frontend and backend changes. Our approach can be adopted to reimplementing other model codes, and provides a good template for porting weather and climate models between languages.

We have shown the performance implications of the DSL design and implementation, leading to a ~3.3x speed up for 460 Pace's GPU backend over the Fortran reference. One advantage of Python is the blending of compile- and run-time allows us to compile the model for a specific runtime configuration. Increasing the amount of code exposed to the DSL compiler had

a strong impact on our model performance and drove us to adopt DaCe to leverage full-program optimizations. Algorithmic changes such as the duo-grid implementation can drive further optimization.

Pace takes great advantage of the Python ecosystem. Pace has full access to Python packages such as Numpy (Harris et al., 2020a) and matplotlib (Hunter, 2007). We can run the model, or just a model component, in a Jupyter notebook, opening new avenues for model exploration and debugging. The Python frontend also allows Pace to easily incorporate and couple to machine-learned model components, such as physics parameterizations. We have also illustrated the relative ease of debugging Pace, both through Python's developer tools and through the ability to implement novel tests like subtracting two dycores to determine whether temporary variables are properly reset between timesteps.

As supercomputer heterogeneity increases, Pace stands as both a useful atmospheric model and a strong proof-of-concept for the DSL approach to performance-portability. We have shown the advantages of an atmospheric model written in a high-level language such as Python, with further development still to come. We hope to see this approach adopted more broadly in the modeling community.

*Code and data availability.* Code releases for Pace (George et al., 2022, https://doi.org/10.5281/zenodo.7079980) and gt4py (Dahm et al., 2022, https://doi.org/10.5281/zenodo.7080260) are available on Zenodo.

*Author contributions.* Johann Dahm, Eddie Davis, Florian Deconinck, Oliver Elbert, Rhea George, Jeremy McGibbon, Tobias Wicky, Elynn Wu: equal contributions to Methodology, Software, Validation, and Writing. Christoper Kung: Software, Validation. Tal Ben-Nun, Linus Groner: Methodology, Software, Validation. Lucas Harris: Validation, Writing. Oliver Fuhrer: Conceptualization, Methodology, Writing, Project administration, Funding acquisition.

*Competing interests.* The authors declare they have no cometing interests.

*Acknowledgements.* We acknowledge the contributions of Mark Cheeseman, Yannick Niedermayr, and Mikael Stellio to the Pace codebase, domain partitioning code, and the port of the microphysics, respectively. We thank Rusty Benson (NOAA/GFDL) for insightful discussions. Further, we acknowledge contributions from the whole GT4Py team, specifically Hannes Vogt (CSCS) and Enrique Gonzalez (CSCS), for their help in implementing a validating version of FV3 using GT4Py. This work was supported by a grant from the Swiss National Supercomputing Centre (CSCS) under project ID s1053. We thank Vulcan Inc., the Allen Institute for Artificial Intelligence (AI2), the Geophysical Fluid Dynamics Laboratory (GFDL) of NOAA, and the Global Modeling and Assimilation Office (GMAO) of NASA for supporting this work. Tal Ben-Nun is supported by the Swiss National Science Foundation (Ambizione Project #185778).

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
