# Peer review of "Pace v0.2: A Python-based Performance-Portable Atmospheric Model"

_EGUsphere, 2022_

## Author Response (AR1)

**Reply on RC1**

We thank reviewer 1 for their thoughtful and constructive comments which have helped improve the manuscript.

Major:

1. F1: Workflow pipeline …

   RC: The pipeline figure is reasonable - however, please include a brief statement on the role of DaCe in this pipeline (either in the text associated with F1, or in the F1 caption). Figure 1 is referred to in line 98, however DaCe is not commented on until lines 125, and Section 5.

   Reply: We have updated the caption and the text discussion of F1 to address the use of DaCe.

2. L12: Pace demonstrates how a high-level language can … facilitate the integration with new technologies such as machine learning.

   RC: Are there currently available demonstrations of machine learning methods being used in Pace? If so, it would be useful to include them in the paper. Figure 13 and the text in Section 6.2 address this to some extent. It is useful to include changes that "could be" (caption Fig 13) made to the model to highlight potential use-cases, however I believe a model development and technical paper must emphasize elements that are currently operational.

   Reply: Pace is not an operational model, since it does not yet include a full suite of physics packages. As a consequence, there are also no operational machine learning use-cases. Nevertheless, the example given in F13 and Section 6.2 is not just theoretical. We have integrated a machine learning component which has been trained using data from the Fortran version of fv3gfs to emulate the Zhao-Carr microphysics into Pace. We have added a figure to illustrate a simulation with this ML emulator based on the code shown in F13 to the paper.

3. F3: RC: Is the model constrained to a specific floating point precision type?  Have results been verified in all supported float-type options?

   Reply: The type FloatField in the code shown in F3 is configurable to different float types (see https://github.com/ai2cm/pace/blob/f5a4848e909339d71ffab0a6f8bdcce5c99e459d/dsl/pace/dsl/typing.py#L29 ). But Pace currently has been only tested using double precision

(float64). The caption of F3 has been extended to explain the former. A sentence stating the latter has been added at the end of Section 4.2 (Model Validation).

4. L205: … we occasionally needed to extend GT4Py …

RC: Is there clarity in when these extensions were/will be needed ? Are these discovered as the code is being ported, or did the design stage make it abundantly clear that certain extension patterns would be necessary? (I.e. do you anticipate additional scenarios where the DSL needs to be extended beyond those highlighted in section 4.1, especially when considering parameterizations?)

Reply: We have added a discussion at the end of Section 4.1. In summary, a DSL will always be restricted to a certain domain and not be able to match all algorithmic motifs that are expressable in a general purpose language such as Fortran. But we believe that the feature set of GT4Py is sufficient to already be interesting for a wide range of applications. GT4Py is currently being extended to be able to express computations on unstructured grids, such as the ones used by the MPAS and ICON models.

Some of the extensions to GT4P were anticipated at the beginning of our work, such as a horizontal regions-type capability, given the nature of the FV3 dynamical core. The addition of while-loops, on the other hand, only became a priority after we measured the performance of the original vertical remapping port.

5. T1: RC: Exactly what is being timed ? Please be more precise in the table caption. (There is mention of the absolute runtime in L267 but the table needs a more appropriate caption).

Reply: Table caption has been revised and the text adapted. The units of the "Time" column erroneously were "ms" instead of "s". This has also been corrected.

6. F10: axis label [time per timestep]

RC: The units appear to be inconsistent with those in Table 1.

Reply: With the correction of "ms" to "s" F10 is now consistent with T1.

7. L314, L351: "certain conditions"

RC: Multiple references to possible optimization based on specific conditions - documentation of such conditions would improve the quality of the paper and enable readers to make decisions on porting strategies if a similar exercise A sd were to be carried out in the future.

Reply: Rephrased section 5.1 to make it clearer that the DSL compiler automatically applies this under certain conditions, as long as it does not change the result of the code.

The nature of the DSL is not to code for these optimizations, but to let the compiler apply them to the extent it can.

8. L333: We leverage DaCe …

RC: See comment (1).

Reply: Addressed in reply to comment (1).

9. L402: "We have preliminary results … competitive performance …"

RC: I believe including these (quantitative) results strengthens the paper by supporting the general statements made on the feasibility of incorporating parameterizations. Include quantitative measures of competitive performance here.

Reply: We feel that our work on the microphysics implementation is no longer best described as preliminary, and have updated the text of the paper to reflect this. We have added discussion of the microphysics performance to section 5, but the execution time is so short (under 1%) that it has a negligible impact on our overall performance.

10. L433-435:

RC: Clarify - is the intent to represent Pace v0.1 as a complete atmosphere model with physical parameterizations or as the underlying dynamical core? The model development paper in review addresses the dynamical core component of a weather / climate model. As such I believe the statement "We have shown the advantages of a climate model written in a high-level language" should be rephrased to emphasize this. Following the code documentation, an alternative way to support the current statement in L434 could be to elaborate on the `fv3gfs-physics` component with supporting results (This is related to lines 401-404).

Reply: We are in the process of integrating the GFS physics suite which is also used in FV3GFS and x-SHIELD. We have a full suite of validated parameterizations ported to GT4Py. Also, we have written a physics driver and already integrated the microphysics parameterization into Pace. Pace will become a full atmospheric model, though Pace v0.1/v0.2 only has microphysics integrated so far. We have edited "climate model" to "atmospheric model" and included additional details about our efforts adding additional parameterizations into the "Limitations" section to reflect this.

12. Plain Language Summary: Similar to (11) - "We re-wrote a Fortran code that simulates weather and climate into Python …" If you are suggesting here that, as submitted, Pace v0.1 is a Python weather and climate simulator then this must be supported by benchmarks in more complex scenarios (with the appropriate physical parameterizations included).

Minor:

1. L14: "Current weather and climate models are written in low-level compiled languages…"

   RC: Please support this with appropriate citations.

   Reply: Reference added.

2. L19: "There [are] a handful of successful … "

   Reply: Corrected.

3. F5: "… [figrue] …" -> "… [figure] …"

   Reply: Corrected.

4. L47: RC: Please include an explicit citation for the FV3 dynamical core. I notice that this is done in line 131, but it is useful to include the citation at the first mention of this in the text.

   Reply: Reference added.

6. L99: RC: Are there any pitfalls when handling "generated" Numpy code?

   Reply: The generated Numpy code is "standard" Python code and can - for example - be executed without special hardware and debugged using a standard Python debugger. Nevertheless, it is important to keep in mind that this is generated code. While GT4Py tries to generate as readable code as possible, some variable names are auto-generated and a direct link between the original source code and the Numpy generated code can be obscured and difficult to understand for a novice user of GT4Py.

7. L119: [horizontal difference of the u and v contravariant velocity components respectively.]

   Reply: Corrected.

8. L203: "… [lanches] …" ->  "… [launches] …"

   Reply: Corrected.

9. L284: [optimization].

   Reply: Corrected.

   RC: The text in this section in general is within the stated scope - the paper demonstrates portability but it does not demonstrate fully optimized CPU portability (authors state this is future work).

Reply: We agree with the reviewer. On the target system we used for performance measurements the memory bandwidth on the GPU (NVIDIA Tesla P100) is approximately 12x higher as compared to the CPU (Intel Xeon E5-2690 v3). This is approximately consistent with the runtime difference of Pace on GPU and CPU. While a more detailed investigation would be required, it could also be argued that we have consistent CPU and GPU performance but should be able to further improve performance as compared to the Fortran reference. But as stated in the text, we are indeed confident that we can improve CPU performance without too much effort.

10. F11: RC: Is this simulated time per timestep? Please clarify the precise measure being timed in the figure caption.

Reply: Corrected.

11. L288: …[across simulation scale]

RC: … [across simulation scales] …

Reply: Corrected.

12. L408: "… as explained in [Section 5] …"

Reply: Corrected.

13. L414: Mixed tenses, please fix.

Reply: Corrected.

Other: Please include Zenodo DOI references to `Pace` and `gt4py` in the code availability section in the manuscript. (The code is otherwise accessible from the "assets" section on the preprint submission page.)

Reply: We have added direct URLs to the Zenodo citations in the code availability section.

**Reply on RC2**

Comment 1: Further discussion of other Python acceleration techniques would help put GT4Py into context. Cython and Numba are mentioned in passing, but why were they not chosen for the FV core? What about JAX? Did the authors try any of these? Were there reasons from the outset that these approaches would be limited relative to the GT4Py approach? More discussion here would be very useful to other authors interested in moving other high-performance codes to Python.

Reply: Addressed in new subsection 2.2.

Comment 2: More discussion of other Python and Julia-based PDE solvers and GFD models would also help contextualize the solver. Popular Python-based (or Python-interfacing) PDE DSLs such as FEniCS, Exasim, and Dedalus might be mentioned. Also other atmosphere and ocean models being implemented in high-level languages on GPUs, such as CliMA and Veros, should be referenced and contrasted. Again I don't think direct simulations comparisons are necessary, but discussing these projects and how the Pace developers see their code in relation would be very helpful to others.

Reply: Addressed in new subsection 2.2

Comment 3: The scaling tests are a little confusing. The exact setup (nodes vs ranks and total model degrees of freedom) are scattered throughout the text, but should be stated clearly in each figure/caption. The weak scaling results look particularly impressive, but again the details aren't clear -- the plot says number of nodes, but the text refers to the left-most point is refereed to as "6 ranks". The text says that the Fortran reference is ran with 6 MPI ranks per node, but doesn't specify this number for Pace. Is this the same, or is it 1?

Reply: Figure 11 (now Figure 10) was updated to refer to Ben-Nun et al. (2022) where full detail of the setup can be found. The description of our mpi configurations at the start of section 5 has been updated to include the MPI configuration for our Pace runs. We have also consolidated the setup where possible to the start of the section, and duplicated the description of node setup into the caption for Figure 11.

Comment 4: Finally, the strong scaling test leaves a lot to be desired and should be expanded. If it's possible to run Pace up to 864 nodes, then it would be much better to see a broader strong scaling test that illustrates the opposing limits of fitting the problem in GPU memory vs. having too little local work for the GPU to do. Understanding the window of local-work-per-node required for maximum performance is very important to potential users, and seeing the efficiency penalties either side of the optimum is also key.

Reply: Since we were able to show perfect weak-scalability we chose to show the sensitivity of the performance with respect to the work per node not by traditionally strong scaling with a fixed experiment size and an increasing number of nodes as this would significantly decrease the observable spectrum. Instead we show a detailed analysis of the interaction between performance and work to be done per node. We added Figure 11 to show where the window of maximal efficiency for GPUs is as well as simplified Table 1 to make it easier to extract the relevant information there. These changes are accompanied by changes to the text to explain this reasoning.